# Z/I1 Hybrid Virulence Plasmids Carrying Antimicrobial Resistance genes in *S.* Typhimurium from Australian Food Animal Production

**DOI:** 10.3390/microorganisms7090299

**Published:** 2019-08-29

**Authors:** Ethan R. Wyrsch, Jane Hawkey, Louise M. Judd, Ruth Haites, Kathryn E. Holt, Steven P. Djordjevic, Helen Billman-Jacobe

**Affiliations:** 1The ithree Institute, University of Technology Sydney, Sydney, NSW 2007, Australia; 2Department of Infectious Diseases, Central Clinical School, Monash University, Melbourne, VIC 3004, Australia; 3Asia Pacific Centre for Animal Health, Melbourne Veterinary School, Faculty of Veterinary and Agricultural Sciences, University of Melbourne, Parkville, VIC 3052, Australia; 4Faculty of Infectious and Tropical Diseases, Department of Infection Biology, London School of Hygiene & Tropical Medicine, London WC1E 7HT, UK

**Keywords:** *Salmonella*, plasmid, antibiotic resistance, AMR, agriculture

## Abstract

Knowledge of mobile genetic elements that capture and disseminate antimicrobial resistance genes between diverse environments, particularly across human–animal boundaries, is key to understanding the role anthropogenic activities have in the evolution of antimicrobial resistance. Plasmids that circulate within the *Enterobacteriaceae* and the Proteobacteria more broadly are well placed to acquire resistance genes sourced from separate niche environments and provide a platform for smaller mobile elements such as IS*26* to assemble these genes into large, complex genomic structures. Here, we characterised two atypical Z/I1 hybrid plasmids, pSTM32-108 and pSTM37-118, hosting antimicrobial resistance and virulence associated genes within endemic pathogen *Salmonella enterica* serovar Typhimurium 1,4,[5],12:i:-, sourced from Australian swine production facilities during 2013. We showed that the plasmids found in *S.* Typhimurium 1,4,[5],12:i:- are close relatives of two plasmids identified from *Escherichia coli* of human and bovine origin in Australia circa 1998. The older plasmids, pO26-CRL_125_ and pO111-CRL_115_, encoded a putative serine protease autotransporter and were host to a complex resistance region composed of a hybrid Tn*21*-Tn*1721* mercury resistance transposon and composite IS*26* transposon Tn*6026*. This gave a broad antimicrobial resistance profile keyed towards first generation antimicrobials used in Australian agriculture but also included a class 1 integron hosting the trimethoprim resistance gene *dfrA5*. Genes encoding resistance to ampicillin, trimethoprim, sulphonamides, streptomycin, aminoglycosides, tetracyclines and mercury were a feature of these plasmids. Phylogenetic analyses showed very little genetic drift in the sequences of these plasmids over the past 15 years; however, some alterations within the complex resistance regions present on each plasmid have led to the loss of various resistance genes, presumably as a result of the activity of IS*26*. These alterations may reflect the specific selective pressures placed on the host strains over time. Our studies suggest that these plasmids and variants of them are endemic in Australian food production systems.

## 1. Introduction

Members of the *Enterobacteriaceae* are important nosocomial and veterinary pathogens capable of causing both intestinal [1] and extraintestinal [2] infections and are often zoonotic [3]. Their universal presence in the gastrointestinal tracts of vertebrates, particularly humans, food animals and avian species, exposes them to diverse antimicrobial selection pressures. Sound antimicrobial stewardship is an important strategy needed to curtail the evolution and spread of multiple-drug-resistant (MDR) pathogens, particularly to prevent or restrict use of clinically important antibiotics in food production systems [4]. Whole genome sequencing and PCR-based studies of commensal and pathogenic *Escherichia coli* and *Salmonella* sourced from intensive livestock production systems in Australia show infrequent and sporadic carriage of genes encoding resistance to antibiotics of critical importance in human medicine [5,6,7,8,9,10,11] and attest to the effectiveness of antimicrobial stewardship practices in Australian food production systems. However, plasmid sequences derived from Australian *E. coli* and *Salmonella* from healthy humans [12] and food animals [13,14,15] show widespread carriage of genes encoding resistance to first generation antimicrobial agents and heavy metals. Carriage of plasmids carrying genes encoding resistance to clinically important antimicrobials is likely to be circulating widely in companion animals and horses [4,16,17], but further studies are needed to address this hypothesis. 

In Australia, IS*26* has played a major role in altering the structure of class 1 integrons [7,9] and the capture and assembly of antimicrobial resistance (AMR) genes on plasmids [8,12,18], chromosomes [19] and transposons [8]. In a previous study, we inferred from sequence data that IS*26* had introduced AMR genes into the backbones of mercury resistance transposons that are mobilised by I1 (pO26-CRL_111_) and related Z (pO26-CRL_125_) plasmids that co-resided in enterohaemorrhagic (EHEC) O26 strain 06877 from a haemorrhagic colitis patient in Victoria, Australia in 1998 [14]. We also described how Z plasmid pO26-CRL_125_ shared a high sequence identity with Z plasmid pO111-CRL_115_ from bovine strain D275, an atypical enteropathogenic (aEPEC) O111 isolate also from the late 1990s [15]. All three plasmids appeared to have acquired identical Tn*21* derivative transposons carrying Tn*6026*. Tn*6026* is an IS*26*-mobilised transposon encoding *bla*_TEM-1_ (ampicillin resistance), *sul2* (sulphonamide resistance), *strAB* (streptomycin resistance) and *aphA1* (neomycin/kanamycin resistance) [8]. pO26-CRL_125_ and pO111-CRL_115_ are near-identical mosaic plasmids, carrying a Z replicon, and comprise conjugal transfer and stability genes that show high sequence identity with I1 plasmids like pO113 (GenBank AY258503). These plasmids are also host to a virulence-associated locus encoding a putative serine protease autotransporter (SPATE), and a complex resistance locus (CRL—here used as a general description for clustered resistance genes and mobilising genetic elements) [15] formed by a Tn*21* mercury resistance transposon, hosting a Tn*402* derivative clinical class 1 integron with a *dfrA5* gene cassette (trimethoprim resistance), interrupted by Tn*6026*. A truncated Tn*1721* module encoding *tetAR* (tetracycline resistance) was at the terminus of the structure, identical to Tn*1721* modules found in P-1-beta plasmid pB10 (GenBank NC_004840) and N plasmid pRSB201 (GenBank JN102341). The truncated Tn*1721* module was separated from the derivate Tn*21* transposon by a region identical to the *oriV* of the P-1α plasmid pBS228 [15]. The Tn*6026* insertion presumably modified the class 1 integron 3′-conserved segment (3′-CS), which comprises only 24 bp. This deletion signature is often, but not exclusively, associated with the insertion of Tn*6026* or related transposon Tn*6029* in MDR *E. coli* from the faeces of Australian agricultural animals [7,9,20]. 

*Salmonella enterica* subspecies *enterica* serovar Typhimurium (*S*. Typhimurium) is a globally distributed serovar implicated in nontyphoidal human and animal salmonellosis, and the dominant serovar causing foodborne human infections in Australia [21]. Recently, a monophasic variant of *S*. Typhimurium with the serotype 1,4,[5],12:i:- and Phage Type (PT) 193 was identified in Australian swine [22]. The increase in frequency of *S*. 1,4,[5],12:i:- PT193 in Australia mirrors the dominance displayed by other monophasic variants of *S.* Typhimurium with the same serotype in Europe, where swine are also a major reservoir [23,24], as well as Asia, Latin America and the USA [25].

Five independent and geographically separate farrow-to-finish swine production operations in Australia were recently shown to be a source of *S*. 1,4,[5],12:i:- PT193. Pigs in four of these facilities showed substantial shedding patterns indicative of a high hazard load in market weight animals [22]. Multiple locus variable number tandem repeat analysis (MLVA) suggested that each of five farms carrying *S*. 1,4,[5],12:i:- PT193 carried an endemic strain or closely related strains represented by 12 closely-related MLVA profiles. Furthermore, MLVA patterns of *S*. 1,4,[5],12:i:- PT193 sourced from human and nonporcine origins collected during Australian surveillance suggest a highly related population may be circulating in Australia.

Here, we describe the complete sequences of two plasmids from *S*. 1,4,[5],12:i:- sourced from commercial swine production facilities in 2013. The plasmids, pSTM32-108 and pSTM37-118, are near identical to IncZ plasmids pO26-CRL_125_ from human strain 06877 and pO111-CRL_115_ from bovine strain D275 (1998) that are described above. We characterised the evolutionary changes these plasmids have undergone, many of which are localised to the CRLs.

## 2. Methods and Materials

### 2.1. Strains

Plasmids pSTM32-108 and pSTM37-118 were sourced from *S.* Typhimurium strains TW-Stm32 and TW-Stm37, respectively. Strain collection methods and typing as *S*. 1,4,[5],12:i:- PT193 have been reported [22]. Briefly, the strains were acquired from the same Australian commercial herd of weaner pigs, sampled at different times during 2013. Sampling was performed by aggregating portions of undisturbed faecal pats taken from the pen floor, followed by the isolation of *Salmonella* [26] from the aggregate samples. TW-Stm37 displayed resistance to tetracycline and trimethoprim, whilst TW-Stm32 displayed resistance to tetracycline, trimethoprim and ampicillin.

### 2.2. DNA Extraction, Sequencing and Genome Assembly

Strains TW-Stm32 and TW-Stm37 were cultured in LB broth at 37 °C for DNA extraction. Genomic DNA from pure isolates was extracted using the JANUS Chemagic automated workstation (PerkinElmer^®^, Massachusetts, United States) with the Chemagic Viral DNA/RNA kit (PerkinElmer^®^, Massachusetts, United States). Unique dual indexed libraries were prepared using the Nextera XT DNA sample preparation kit (Illumina^®^, San Diego, United States). Libraries were sequenced on the Illumina NextSeq^®^ (Illumina^®^, San Diego, United States) 500 with 150-cycle paired end chemistry as described by the manufacturer’s protocols.

Long read sequencing was performed using the Oxford Nanopore Technologies (ONT) MinION Nanopore (Oxford Nanopore Technologies^®^, Oxford, United Kingdom) using LSK-108, EXP NBD-113 chemistry and sequenced on an R9.5 flow cell. For Illumina, average chromosomal read depth was 64x and 60x for TW-Stm32 and TW-Stm37, respectively, with an average read length of 151 bp. For ONT, the N50 read length was 7690 bp and 13,957 bp for TW-Stm32 and TW-Stm37, respectively. Reads were filtered using Filtlong (https://github.com/rrwick/Filtlong) to keep only the best reads up to a target of 500,000 bp, and to remove any reads shorter than 1000 bp.

Illumina and filtered ONT reads were combined to generate hybrid whole genome assemblies using Unicycler v0.4.1 [27] with default parameters. Both plasmids were complete and circularised and extracted from the resulting assembly graph.

Sequences have been deposited in GenBank under the accession numbers MN334219 and MN334220.

Gene annotation was performed using NCBI BLASTn server and reference plasmids pO26-CRL_125_ (KC340960.1) and pO111-CRL_115_ (KC340959.1). Annotations were managed with SnapGene. Contextual sequence comparisons were generated with progressiveMauve [28]. Plasmid replicon typing was confirmed using PlasmidFinder [29]. Plasmid MLST was performed using the service provided by the Centre for Genomic Epidemiology (CGE) [29]. Detection and comparison of single nucleotide polymorphisms was performed with Parsnp v1.1 [30], utilising the PhiPack recombination filter. Parsnp selects a conserved core based on homology between all input sequences. Insertion sequences were identified with ISfinder [31]. Figures were generated using SnapGene and BRIG [32].

## 3. Results 

Despite being isolated approximately 15 years apart, pSTM32-108 (108,065 bp) and pSTM37-118 (118,115 bp) are near identical to Z plasmids pO26-CRL_125_ and pO111-CRL_115_ (Figure 1). Plasmids pSTM32-108 and pSTM37-118 carry *repZY* from Z plasmids, detectable by PlasmidFinder as a B/O/K/Z plasmid at 95% identity. Plasmid MLST utilising the I1 pMLST scheme [33] identifies only the *ardA_*2 allele as part of the I1 plasmid backbone present on each plasmid.

Using pO26-CRL_125_ as reference, core polymorphism analysis performed with Parsnp identified only one variant site in the conserved backbone of pO26-CRL_125_, while a separate single variation was also detected in pO111-CRL_115_ (Figure 2). One unique site was also shared between pSTM32-108 and pSTM37-118, suggesting neither of the earlier plasmids was the direct originator of pSTM32-108 and pSTM37-118, but that they all share a common ancestor. The SNPs identified were in noncoding sequence of the *par/imp* region in pO111-CRL_115_, in a DNA modification methylase within pO26-CRL_125_, and in a putative transcriptional regulator within the virulence locus of pSTM32-108 and pSTM37-118. The core sequence selected by Parsnp covered between 73% (pO26-CRL_125_) and 85% (pSTM32-108) of each plasmid sequence, including homologous regions of the resistance structure (Figure 2).

Sequence comparisons show that modifications to the later plasmids were largely due to the activity of mobile genetic elements. Plasmids pSTM32-108 and pSTM37-118 acquired an IS*66* family element (most similar to IS*Cro1*), indicative of an insertion into a common ancestor plasmid. The IS66 family element is flanked by 8 bp direct repeats (GTTTGTTT) and situated between the B plasmid region and the *trbCBA* operon (Figure 1), interrupting a hypothetical coding sequence (CDS) with putative conjugative function. The I-associated module encoding proteins with putative roles in plasmid partitioning and segregation in pO26-CRL_125_ are also present in pSTM32-108 and pSTM37-118. Venturini et al. (2013) [15] identified recombination signatures both bordering and within this module, with one copy of this signature present at the recombination site in pO111-CRL_125_. Identical signatures were observed in pSTM32-108 and pSTM37-118. Plasmid pO26-CRL_125_ was also missing 135 bp of *traH*, which pSTM32-108 and pSTM37-118 retained. This portion of *traH* was excluded from SNP analysis (data not shown), as pO26-CRL_125_ was the reference sequence for data presented.

While pSTM32-108 and pSTM37-118 are host to the same derivative Tn*21* transposon characterised in earlier studies [14,15], insertion and deletion events seemingly mediated by IS*26* have led to changes in the CRL in each plasmid (Figure 3). Initially, a precursor plasmid to pSTM32-108 and pSTM37-118 likely had an IS*26* insertion downstream from the *dfrA5* cassette, 709 bp past *intI1*, with 5 bases now separating the end of the cassette and the terminal inverted repeat of IS*26*, disrupting the cassette *attC* signature. In pSTM37-118, the *tniA* CDS of Tn*21* directly follows the IS*26* element at this new locus, giving a structure with the entire 3´-CS and Tn*6026* lost (Figure 3). This effectively removes *bla*_TEM_, *strAB, sul2* and *aphA* from the plasmid resistance profile, leaving the plasmid with only resistance to trimethoprim (*dfrA5*), tetracycline (*tetAR*) and mercury (*mer*).

In pSTM32-108, the IS*26* near *dfrA5* flanks the Tn*6026 bla*_TEM_ module, now inverted compared to pO26-CRL_125_. Conserved plasmid sequence then follows the *bla*_TEM_ module; however, 327 bp of this sequence has also been lost, assumedly during these alterations. With this, the majority of sequence from Tn*6026*, Tn*21*/*mer* and Tn*1721* has been lost. This leaves pSTM32-108 encoding only *dfrA5* and *bla*_TEM_. The loss of these transposable units led to the structures observed; however, this loss also removed the genetic signatures necessary to determine the precise evolutionary path of these diverged resistance structures. Nonetheless, a plausible evolutionary path using two key IS*26* insertions is outlined in Figure 4. Depicted are two intermediate IS*26* insertion events. The first is common to both later structures, an insertion event moving the IS*26* border nineteen bases closer toward *intI1*. The loss of *Tn6026* would then lead to the structure in pSTM37-118. Second is an IS*26* insertion into the plasmid backbone, past *tnpA_1721_*, leading to the eventual formation of the structure in pSTM32-108.

## 4. Discussion

The swine herd from which *S*. Typhimurium strains TW-Stm32 and TW-Stm37 were sourced was one of five sampled in the study by Weaver et al. (2017) [22]. Isolates recovered from the herd in 2013 displayed a combination of resistances to tetracycline, trimethoprim, ampicillin, streptomycin and sulphonamides. Strain TW-Stm37 specifically displayed resistance to tetracycline and trimethoprim, and TW-Stm32 displayed resistance to tetracycline, trimethoprim and ampicillin. These profiles are reflected by the gene content identified in plasmids described here, aside from the tetracycline resistance observed in TW-Stm32, hosted chromosomally. 

Genetic elements such as IS*26* have been linked with the formation of atypical and complex class 1 integron structures [7,20,34], independently mobile compound transposons [8,15] and complex resistance gene loci (CRLs) [12,14,15,18,19,35]. We hypothesised that the continued use of first-generation antimicrobials in farm animal production systems shapes the genetic structures of plasmids and chromosomal islands carried by commensal and pathogen populations of *Enterobacteriaceae* such that they not only acquire resistance to these antimicrobials but display an enhanced ability to acquire further resistance with a reduced fitness cost imposed on the host. The insertion sequence IS*26* is significant in this regard because of its propensity to: (i) mobilise resistance genes belonging to most classes of antibiotics [36,37]; (ii) generate random deletions next to their site of insertion [7,9,38,39]; (iii) shuffle gene content [19,35]; (iv) create hybrid plasmids encoding combinations of virulence and resistance genes [14,15,18,40]; (v) create independently mobile, IS*26*-flanked transposons [8,14,37,41] which influence antibiotic resistance gene carriage in chromosomal islands [19,35]; and (vi) remove genes that negatively influence fitness [42].

With the activity of insertion sequences being the major differentiating factors between such closely related plasmids as the four discussed here, tracing them using molecular diagnostic tools requires the identification of unique DNA signatures. One potential example observed here is the insertion which brought IS*26* in closer proximity to *intI1*. IS*26* can also act as a border for the site-specific insertion and removal of other genes mobilised by IS*26* and is one reason the epidemiological analysis of these elements is crucial to tracing AMR evolution. While IS*26* activity led to the loss of various resistances here, its actions can lead to the stable acquisition of future DNA just as readily. Detailed information on specific insertion sites will assist in untangling the developmental path of any future isolations.

An interesting factor in the development of both *S.* Typhimurium plasmids here is the loss of sulphonamide resistance. Sulphonamide drugs are amongst the earliest antibiotics used to treat infection, and consequently, the resistance genes *sul1, sul2* and *sul3* have been consistently observed in association with class 1 integrons and the spread of AMR [43,44,45,46]. The loss of *sul* genes here suggests that pSTM37-118 and pSTM32-108 have potentially developed niche-specific resistance profiles, although previous survey data have shown sulphonamides to be used widely in Australian porcine agriculture in general [47], so this may simply be an artefact of the two samples available.

Plasmids like those described here are a primary route for the dissemination of virulence and AMR mechanisms amongst commensal and pathogenic *Enterobacteriaceae.* The alteration and consolidation of resistance gene cargo within transferrable resistance loci, often associated with the class 1 integrase, leads to diverse, complex resistance gene structures. The challenge in Australia is to monitor how these plasmids move between food animal production systems, humans, the environment and agriculture more broadly. These examples add to a growing number of reports of MDR plasmid sharing between *E. coli*, *Salmonella enterica* and other members of the *Enterobacteriaceae* [48,49,50,51]. Genomic surveillance efforts are in their infancy in Australia, and it is likely that many more instances of plasmid sharing within the *Enterobacteriaceae* and perhaps other bacterial genera will become evident as surveillance becomes routine and metagenomic methodologies become more sophisticated [52,53]. At present, the Z/I1 hybrid plasmids described here are limited to these isolations from Australian *Enterobacteriaceae*, and any inferences we make may only be pertinent to these specific examples; however, it is likely variants of these plasmids are widely disseminated given the length of time since they were isolated both here, and from cattle and human sources [14,15]. A routine genomic epidemiological surveillance strategy is needed globally to determine the identities of plasmids and the genetic cargo they carry and understand their mobility and the contribution they make to the spread AMR and pathogen evolution.

## Figures and Tables

**Figure 1 microorganisms-07-00299-f001:**
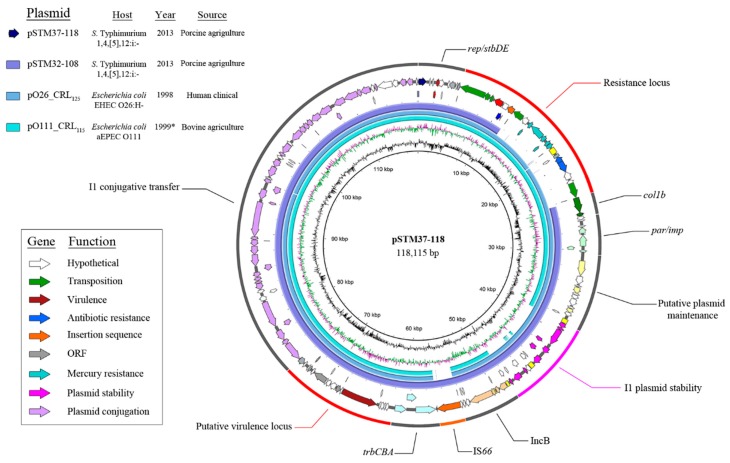
Map of IncZ/IncI derivative plasmid pSTM37-118 alongside BLASTn alignments of related plasmids Annotated map of pSTM37-118 and a table of details describing plasmids included in the analysis. Colour-coded BLASTn alignments of pSTM32-108, pO26-CRL_125_ and pO111-CRL_115_ are shown in the inner circles. Inner graphs show GC skew (purple/green) and GC content (black) of pSTM37-118. Regions of the plasmid are labelled on the outer circle.

**Figure 2 microorganisms-07-00299-f002:**
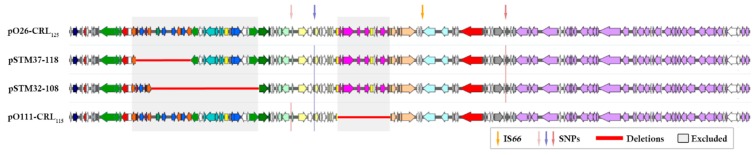
Schematic alignment of plasmids. Comparisons of plasmid annotation schematics, based on pO26-CRL_125_ of highlighted regions, were excluded from the SNP analysis based on a lack of homology. Arrows indicate SNP variant sites and the IS*66* family insertion site, and red bars indicate sequence lost form the individual plasmids.

**Figure 3 microorganisms-07-00299-f003:**
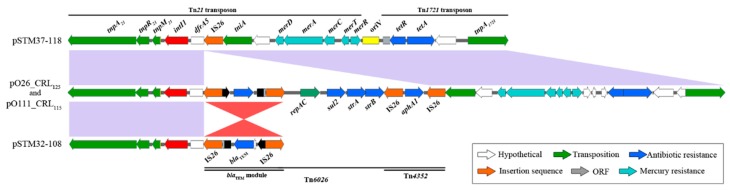
Comparison of resistance-encoding structures formed from Tn*21* and Tn*1721*. Resistance structures present in the four plasmids, demonstrating a loss of sequence in pSTM37-118 and pSTM32-108. Homologous regions between structures are represented in purple and red, with red indicating the IS*26*-mediated inversion of sequence.

**Figure 4 microorganisms-07-00299-f004:**
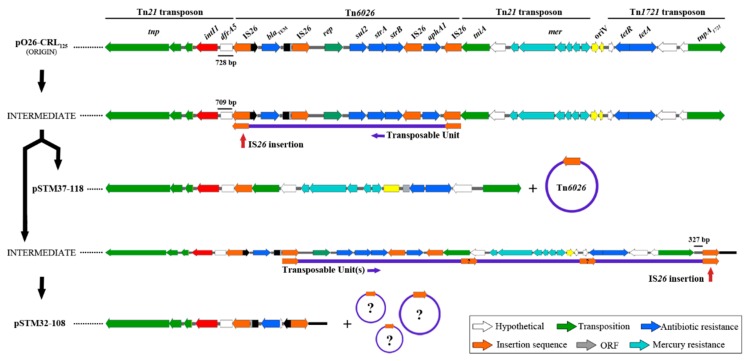
Key IS*26* insertions in the potential evolutionary path of the complex resistance structures Pathway diagram of the transition from the structure sourced in from 1998 [15] (“Origin”) to those present in plasmids isolated from *S.* Typhimurium in 2013. Close under the transitional structures are the potential transposable units (of unknown size and number) which are lost to generate each final complex resistance locus (CRL).

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
