# Peer review of "Z/I1 Hybrid Virulence Plasmids Carrying Antimicrobial Resistance genes in S. Typhimurium from Australian Food Animal Production"

_microorganisms, 2019, doi:10.3390/microorganisms7090299_

Round 1

Reviewer 1 Report

  This paper examines two plasmids encoding AMR and virulence related genes associated with Salmonella originating from pigs in Australia, and thus of agricultural significance. They used long read sequencing to resolve the plasmid structures with a high resolution to discover a high degree of conservation of these plasmids over time, suggesting that scanning prospectively for additional bacteria would be useful. In particular, understanding better the roles of repetitive MGEs like IS26 in rearranged plasmid regions and manipulating AMR genes is relevant to a wider readership. The paper is written very clearly, so the suggestions below are just really minute.   Minor suggestions: - Explain the CRL abbreviation complex resistance gene at the start - Add the GenBank accession numbers - The "challenge" (Discussion) is not just for Australia, it is global

Author Response

Greetings,

As Reviewer’s 1 and 2 had more minor suggestions, changes addressing all reviewers’ comments are combined below. Altered sections have been highlighted in the returned manuscript. We have attempted to respond to most reviewer requests and given reasons otherwise.

Thank you for the responses and suggested edits, they were of great help and useful for consideration.

Regards,

Steven Djorjedvic

Unrequested changes

Line 272 – An additional funding line has been added.

Some minor edits/fixes.

Requested changes

Lines 39-40 – Three references added.

Lines 48-51 – Request to leave as is, we felt re-wording reduced impact.

Lines 54-77 – While we regret the level of detail presented here, this is a description of the data previously reported, related to this study, and request again it be left as is.

Line 70 – Definition for CRL added

Lines 104-106 – Additional sampling description and an additional reference for the microbiology has been added.

Line 122 – Accessions have been added.

Line 205 – references added.

Lines 246, 257-274 – Parts of the discussion re-worded to highlight the limitations of these analyses.

Line 261 – Changed to global impact.

Fixed abbreviations and made edits to remove unnecessary use (line 170)

Given the limited knowledge of these plasmids, we feel commenting on clinical significance is premature.

Reviewer 2 Report

Brief summary

This is a study that characterizes two Z/I1 atypical hybrid plasmids hosting AMR and virulence associated genes within endemic Salmonella enterica serovar Typhimurium 1,4,[5],12:i:- isolates sourced from Australian swine production facilities during 2013. The authors found some alterations within complex resistance regions present on each plasmid that have led to the loss of various resistance genes. The study is limited by the unavailability of a broad sample of swine herds tested. However, this study adds to the current literature on endemic plasmid variants conferring AMR genes in E. coli and Salmonella spp. in Australian food production systems.

Broad comments

This an interesting study that demonstrates the dissemination of AMR and virulence genes across human-animal boundaries that emphasizes the importance of antimicrobial stewardship programs not only in healthcare systems but also in agriculture and veterinary medicine. The study is well-designed and the manuscript is clearly written with a few editing flaws. The “Introduction” is reasonable, given the premise of this study, although it could have been more condensed. The “Methods” and “Discussion” sections are well written but we believe that some additions / clarifications are needed, as explained below. Figures are comprehensive and helpful. This study sets the grounds for further research in AMR dissemination through the food chain.

Specific comments

Major comments

Introduction

In general, introduction and especially the section about IS26 (lines 54-77), needs to be condensed so that it is less confusing for the reader. Parts of this section may be included at the “Discussion” section. Lines 39-40: Please add a reference at this sentence. Lines 48-51: These sentences can be combined to send the same message i.e. the widespread carriage of resistance genes to first generation antimicrobial agents and heavy metals in humans and food animals.

Methods

Line 102-103: Please specify the number of weaner animals that the S. Typhimurium TW-Stm32 and TW-Stm37 strains were sampled from, as well as the number of TW-Stm32 and TW-Stm37 strains analyzed in this study. We suggest that the sampling of the strains, although referenced, should be briefly but precisely described in the “Methods” section of this manuscript. Line 122: Please give the accession numbers of the plasmids’ sequences.

Discussion

Lines 226-234: Please consider rephrasing so that it is clear to the reader that this paragraph expresses a hypothesis based on authors’ findings rather than a proof – of – concept. We agree with the authors that further study of these events, especially if consistent changes are observed, may serve as specific DNA signatures but this hypothesis needs to be confirmed with more sizable studies. Lines 238-241: Again the “niche-specific resistance profiles” is a hypothesis. Could you please elaborate/comment on that? In “Discussion” please include a “limitations of the study” section. We suggest that potential selection/sampling bias as well as other potential limitations and their impact on the results (e.g. the date of the S. Typhimurium TW-Stm32 and TW-Stm37 strain sampling was 6 years ago and the human and bovine plasmids 21 years ago) should also be included in this section.

Minor comments

Please change the placement of the references’ numbers in the text so that they are in agreement with the instructions for authors (“placed before the punctuation”). Please explain all the abbreviations in the text to facilitate the readers. For example: “CRL” is explained in Line 215 while it has been used many times in the text before; “CDS” and “ORF” have not been defined at the manuscript. In Line 50 please review the punctuation (change “;” to “,”). Please give a numbered reference in Lines 205-206 ([19]). Please delete the second “doi:” from the first reference. Please review the reference format. References should have an abbreviated journal title.

Author Response

(The authors gave the same response as above.)

Reviewer 3 Report

Thank you for the opportunity to review this interesting and important manuscript. I believe that the manuscript has been well-written and well-presented. I only have minor comments for the authors to consider - it would be good if the authors can add a few more lines in their discussion on how their findings might be significant clinically

Author Response

(The authors gave the same response as above.)
